# Mixture of Horizons in Action Chunking

**Dong Jing** [1 2]  **Gang Wang** [3]  **Jiaqi Liu** [2]  **Weiliang Tang** [3]  **Zelong Sun** [1]
**Yunchao Yao** [2]  **Zhenyu Wei** [2]  **Yunhui Liu** [3]  **Zhiwu Lu** [* 1]  **Mingyu Ding** [* 2]

## Abstract

Vision-language-action models exhibit an inherent trade-off in action chunk length ("horizon"): longer horizons improve global foresight but degrade fine-grained local control, while shorter ones yield the opposite. To mitigate the trade-off, we propose a **mixture of horizons (MoH)** strategy. In brief, MoH rearranges the action chunk into several segments with different horizons, processes them in parallel with a shared action transformer, and fuses outputs with a light linear gate. It offers three appealing benefits. i) Long-term foresight and short-term precision are jointly exploited within a single model. ii) MoH is plug-and-play for full-attention action modules with minimal training or inference overhead. iii) MoH enables dynamic inference with adaptive horizons, which selects stable actions through cross-horizon consensus, achieving $2.5\times$ higher throughput than baselines while preserving superior performance. Extensive experiments over flow-based and one-step regression policies demonstrate that MoH yields consistent and significant gains on both simulations and real-world tasks.

## 1. Introduction

Vision-language-action models (VLAs) (Kim et al., 2024; Black et al., 2024) have recently attracted increasing attention for their remarkable ability to follow human instructions and execute complex robotic tasks, such as cloth folding (Zheng et al., 2025b), object arrangement (Bu et al., 2025a), beverage preparation (Contributors, 2025), and self-driving (Jiang et al., 2025). VLAs (Kim et al., 2025; Li et al., 2024; Black et al., 2024) enjoy the benefits of large-scale pre-trained vision-language models (VLMs) (Achiam et al.,

[*]Corresponding authors. [1]Renmin University of China [2]University of North Carolina at Chapel Hill [3]The Chinese University of Hong Kong. Correspondence to: Zhiwu Lu <luzhiwu@ruc.edu.cn>, Mingyu Ding <md@cs.unc.edu>.

*Proceedings of the 43$^{rd}$ International Conference on Machine Learning*, Seoul, South Korea. PMLR 306, 2026. Copyright 2026 by the author(s).

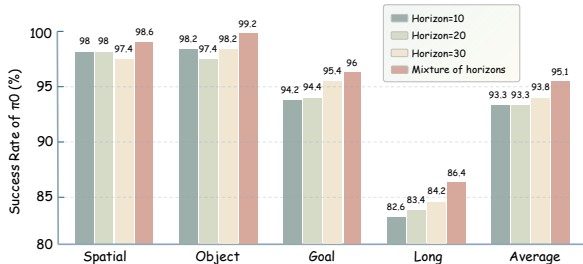

*Figure 1.* Effect of action horizon on $\pi_0$. The first 5 actions in the predicted chunk are executed at evaluation. Varying horizons lead to trade-off effects across four LIBERO task suites. Our mixture of horizons strategy alleviates this trade-off and raises overall success.

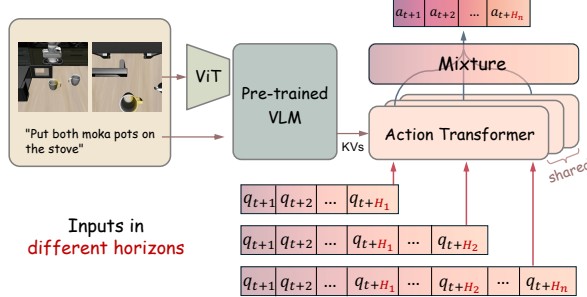

*Figure 2.* Overview of the proposed mixture of horizons strategy, which integrates action chunks of multiple horizons via the shared action transformer and a mixture gating mechanism.

2023; Bai et al., 2025b; Beyer et al., 2024; Zhu et al., 2025), on top of which they are built and further equipped with an action module for embodied control. Typically, modern VLAs adopt this action module with full attention and an ***action chunking*** strategy, allowing the model to predict a sequence of future actions conditioned on the current observation and instruction. Although action chunking has proven effective (Zhao et al., 2023; Shukor et al., 2025), the model's performance is highly sensitive to the chunk length during training, which we refer to as ***horizon***, specifying the temporal span of future actions being predicted. How to understand the effect of horizon selection and its underlying mechanism thus becomes a key problem.

Existing works generally adopt a fixed horizon, which potentially leads to suboptimal performance and limits the model's flexibility, e.g., adaptive control over latency at inference time. To fill this gap, we conduct a study on the influence of horizon. Using the widely adopted $\pi_0$ (Black et al.,

2024) as a baseline, we evaluate horizons $\in [10, 20, 30]$ on the LIBERO (Liu et al., 2023) benchmark. As shown in Figure 1 where tasks ranging from Spatial, Object, Goal, to Long require progressively longer trajectories, we uncover a fundamental trade-off: i) longer horizons improve long-term planning hence better performance on long-horizon tasks, and ii) shorter horizons allow precise control and yield higher success rates on short-horizon tasks. This sensitivity highlights a critical limitation: *a fixed, single horizon imposes an inherent bottleneck on generalization.*

This raises a natural question: *Can we integrate multiple horizons to jointly exploit long-term foresight and short-term precision within a single model?* Motivated by this idea, we propose a **mixture of horizons (MoH)** strategy for action chunking. As illustrated in Figure 2, MoH rearranges each action chunk into multiple segments with different horizons, processes them in parallel using a shared action transformer, and fuses their predictions at each timestep via a lightweight linear gating layer with only $2k$ additional parameters. Incorporating horizons $[10, 20, 30]$ through MoH effectively mitigates the trade-off observed in $\pi_0$ and yields consistent gains across all task suites (Figure 1).

MoH is universal and computationally efficient. It can be seamlessly plugged into any full-attention action module, regardless of whether it is flow-based or single-step prediction-based. In practice, existing action transformers are typically lightweight (around 300M parameters or fewer) when compared with the VLM backbones (Zheng et al., 2025b; Black et al., 2024) and benefit from tensor parallelism, MoH introduces minimal training and inference overhead.

Moreover, MoH naturally enables a dynamic inference scheme via *cross-horizon consensus*. At each timestep, the model outputs horizon-wise predictions together with their mixture. We treat each horizon as a voter and identify actions that receive consistent support across horizons, forming a self-truncating executable chunk while deferring uncertain actions to the next replanning iteration. We demonstrate that this dynamic inference mechanism via cross-horizon consensus improves both execution stability and inference-time efficiency, e.g., even at 2.5× throughput, $\pi_{0.5}$ with MoH still surpasses the performance of the baseline $\pi_{0.5}$.

We evaluate MoH on flow-based models ($\pi_0$ (Black et al., 2024), $\pi_{0.5}$ (Shi et al., 2025) and StarVLA (StarVLA-Community, 2026)) and a single-step prediction model ($\pi_{reg}$ (Black et al., 2024)) across multiple simulation environments and real-world robotic tasks. Across all settings, MoH yields consistent and substantial improvements. Remarkably, under the mixed-task training setting, $\pi_{0.5}$ with MoH achieves an average success rate of **99%** on LIBERO (Liu et al., 2023) after only $30k$ iterations, establishing a new state of the art. Additional ablations and visualizations validate the effectiveness of each component of MoH.

Our contributions are threefold:

1. We present a systematic study of the single action chunking horizon in VLAs, revealing a key trade-off between long-term foresight and short-term precision.
2. We introduce Mixture of Horizons, a plug-and-play, low-overhead approach that alleviates the above trade-off and improves performance and generalization.
3. We propose a dynamic inference scheme via cross-horizon consensus for faster and more stable execution.

## 2. Related Work

**Vision-Language-Action Models.** VLA models (Chen et al., 2024; Fan et al., 2025; Cui et al., 2025; Cen et al., 2025; Zheng et al., 2024; Qu et al., 2025; Deng et al., 2025; Zhong et al., 2025; Zhang et al., 2025a; Wang et al., 2025b; Tang et al., 2025; Li et al., 2025b) map visual observations and language instructions to executable actions for robotic manipulation. Early policy architectures, such as Diffusion Policy (Chi et al., 2023), typically employ relatively small networks and are designed for task-specific scenarios, achieving strong performance but limited generalization. With the rapid progress of VLMs(Achiam et al., 2023; Zhu et al., 2023; Yang et al., 2024; Beyer et al., 2024; Zhu et al., 2025; Jing et al., 2024), recent approaches move toward more general-purpose embodied agents by coupling powerful VLM backbones with action heads or expert modules. For example, OpenVLA (Kim et al., 2024) pretrained a VLA model on large-scale robotic datasets via discrete action token prediction. Recently, the $\pi$-series (Black et al., 2024; Shi et al., 2025) and related methods (Li et al., 2024) adopt flow-matching (Lipman et al., 2022) or diffusion-based policies to predict continuous actions, and have become a dominant design choice. Besides, many works attempt to improve spatial perception (Qu et al., 2025; Lin et al., 2025; Li et al., 2025a; Zhang et al., 2025b) or the cross-embodied generalization (Zheng et al., 2025b; Cheang et al., 2025; Zheng et al., 2025a). VLA models represent a promising pathway toward scalable embodied superintelligence.

**Action Chunking.** Action chunking, popularized by ACT (Zhao et al., 2023), allows policies to predict a sequence of future actions at each control step instead of a single next action. This design exposes the policy to temporal structure, supports high-frequency control, and enables smoother execution by fusing overlapping actions of the same timestamps in different chunks. CogACT (Li et al., 2024) further refines this idea with similarity-based weighting schemes. Consequently, chunked prediction combined with full-attention transformers over the action dimension has become a standard component in modern VLA policies (Kim et al., 2025; Black et al., 2024; Bu et al., 2025a; Gao et al., 2025; Bharadhwaj et al., 2023; Contributors, 2025; Zhai et al., 2025).

Despite its widespread use, the chunk length, termed *horizon*, is typically chosen heuristically. Existing findings (Shukor et al., 2025; Li et al., 2024) suggest that performance is highly sensitive to this horizon and that different horizons are preferable for different task types. Moreover, prior works do not provide an available method to mitigate the trade-off between long-term foresight and short-term precision induced by a fixed horizon. In this paper, we aim to solve this problem by introducing a universal mixture-of-horizons training strategy.

## 3. Method

### 3.1. Preliminaries

**VLA models with action chunking.** VLA models are sequential decision policies for end-to-end robotic manipulation. At each decision step $t$, the policy observes a multi-view input $V_t = \{v_t^{(m)}\}_{m=1}^M$, an optional history $h_{<t} = V_{t-k:t-1}$, a language instruction $T$, and an optional proprioceptive state $s_t$. Instead of predicting a single action, the policy outputs an *action chunk* of length $H$:

$$A_t = (a_{t,1}, \ldots, a_{t,H}) = (a_t, \ldots, a_{t+H-1}) \in \mathbb{R}^{H \times d_a},$$
(1)

where $a_{t,k} = a_{t+k-1} \in \mathbb{R}^{d_a}$ denotes the action at relative step $k$ within the chunk. Action chunking reduces the number of policy calls at test time and enables planning over a temporally extended horizon.

Recent advanced VLA models are typically built upon a pretrained VLM backbone that encodes $(V_t, h_{<t}, T, s_t)$ into a context representation, followed by a compact *action transformer* operating on action tokens. In most cases, the *full-attention* mechanism, where all action tokens attend to each other, is adopted. Many prior works (Zhao et al., 2025; Kim et al., 2025; Black et al., 2024) have demonstrated that this non-causal design consistently outperforms strictly autoregressive decoding for chunk prediction.

**Flow-matching policies.** Flow-matching policies learn a velocity field that transports a Gaussian noise chunk to the target action chunk. Let $\epsilon \sim \mathcal{N}(0, I)$ be a noise chunk of the same shape as $A_t$, and let $\tau \in [0, 1]$ denote a continuous time variable. A standard reference path is the linear interpolation

$$A_t^{(\tau)} = (1 - \tau)\, \epsilon + \tau\, A_t,$$
(2)

whose ground-truth velocity is

$$u(\epsilon, A_t) = \frac{d}{d\tau} A_t^{(\tau)} = A_t - \epsilon.$$
(3)

The flow-matching policy $v_\theta$ is trained to approximate this velocity via

$$L_{\text{fm}}(\theta) = \mathbb{E}_{\epsilon, \tau} \big\| v_\theta\big(A_t^{(\tau)}, \tau, V_t, h_{<t}, T, s_t\big) - u(\epsilon, A_t)\big\|_2^2.$$
(4)

At inference, an ODE solver (Lu et al., 2022) integrates the learned velocity field from $\tau = 0$ to $\tau = 1$ starting from $A_t^{(0)} = \epsilon$ with step size $\Delta\tau$:

$$A_t^{(\tau + \Delta\tau)} = A_t^{(\tau)} + v_\theta\big(A_t^{(\tau)}, \tau, V_t, h_{<t}, T, s_t\big) \Delta\tau, \quad (5)$$

yielding $A_t^{(1)}$ as the final action chunk.

**One-step policies.** One-step policies directly map the context to the final action chunk in a single forward pass:

$$\hat{A}_t = g_\theta(V_t, h_{<t}, T, s_t).$$
(6)

They can be instantiated in either discretized classification or continuous regression form. *(i) Discretized classification.* Each scalar action dimension is quantized into $B$ bins. Let $y_{k,d} \in \{1, \ldots, B\}$ be the target bin index for dimension $d$ of step $k$, and let $p_\theta(y_{k,d} \mid V_t, h_{<t}, T, s_t)$ be the predicted categorical distribution. The loss is

$$L_{\text{cls}}(\theta) = -\sum_{k=1}^{H} \sum_{d=1}^{d_a} \log p_\theta(y_{k,d} \mid V_t, h_{<t}, T, s_t).$$
(7)

*(ii) Continuous regression.* Alternatively, the policy directly regresses continuous actions, e.g., with an $\ell_1$ loss:

$$L_{\text{reg}}(\theta) = \sum_{k=1}^{H} \sum_{d=1}^{d_a} \big|\hat{A}_{t,k,d} - A_{t,k,d}\big|.$$
(8)

Both flow-matching and one-step policies can be built on the basic full-attention action transformer with minor modifications. Specifically, flow-matching requires a time embedding layer, while one-step prediction can be implemented by introducing a learnable query token.

### 3.2. Mixture of Horizons

**Motivation.** As shown in Figure 1, training with a single chunk horizon leads to a trade-off: short horizons favor precise short-term control but lack foresight, whereas long horizons capture long-term structure but may sacrifice immediate motor accuracy. Our goal is to fuse multiple horizons within a single policy so that it inherits the strengths of both.

**Action chunk rearrangement.** We fix a maximum horizon $H$ and a set of candidate horizons $\mathcal{H} = \{h_1, \ldots, h_N\}$ with $h_1 < \cdots < h_N = H$. Given a ground-truth chunk

$$A_t = (a_{t,1}, \ldots, a_{t,H}),$$
(9)

we construct for each $h \in \mathcal{H}$ a truncated chunk

$$A_t^{(h)} = (a_{t,1}, \ldots, a_{t,h}) \in \mathbb{R}^{h \times d_a}.$$
(10)

During training, all horizons share the same observation context $(V_t, h_{<t}, T, s_t)$ processed by VLM. For efficient

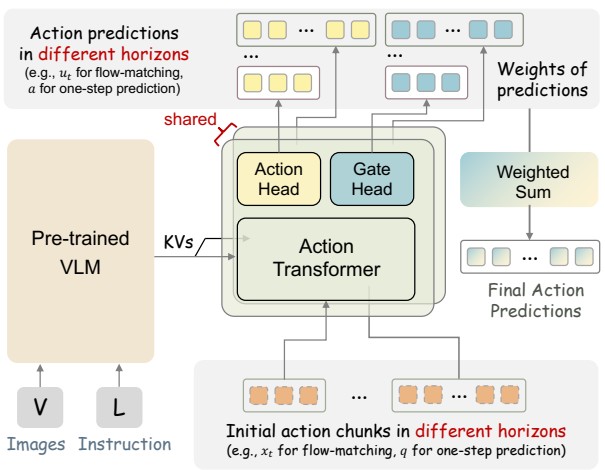

*Figure 3.* Overview of our mixture of horizons framework. The action-related input is rearranged into different horizons and then processed in parallel by a shared action transformer. A linear gate head, with only $2k$ parameters, produces per-step, per-horizon weights to fuse horizon-wise predictions into the final action predictions. This strategy is plug-and-play for any full-attention action transformer, including both flow-matching and one-step policies.

computation, we pad each $A_t^{(h)}$ to length $H$ for batching and use a horizon-specific attention mask that invalidates positions $k > h$. This allows the shared action transformer to process all horizons in parallel in one forward pass. Since the VLM prefix is only computed once and the action transformer is lightweight, our MoH strategy adds negligible computational overhead in both training and inference.

**Gated Mixture.** The shared action transformer produces hidden states $Z_t^{(h)} \in \mathbb{R}^{h \times d}$ for each horizon $h \in \mathcal{H}$. An action head converts these into horizon-specific predictions

$$\hat{A}_t^{(h)} = (\hat{a}_{t,1}^{(h)}, \ldots, \hat{a}_{t,h}^{(h)}), \quad \hat{a}_{t,k}^{(h)} \in \mathbb{R}^{d_a}. \quad (11)$$

As illustrated in Figure 3, following Occam's razor principle (Blumer et al., 1987), we adopt the simplest effective design for fusing horizons: a linear layer is added on top of the shared action transformer as a gating head, which produces logits $g_{t,k,h}$ for each step $k$ and horizon $h$. For a given step $k$, only horizons with $k \leq h$ are valid; we mask out invalid horizons and normalize over the remaining ones:

$$\alpha_{t,k,h} = \frac{\exp(g_{t,k,h})}{\sum_{h' \in \mathcal{H}:k \leq h'} \exp(g_{t,k,h'})}, \quad h \in \mathcal{H}, \, k \leq h. \quad (12)$$

The final fused prediction at step $k$ is

$$\hat{a}_{t,k} = \sum_{h \in \mathcal{H}:k \leq h} \alpha_{t,k,h} \, \hat{a}_{t,k}^{(h)}. \quad (13)$$

This simplest gating leaves the backbone unchanged and applies to any full-attention action transformer, seamlessly integrating with both flow-matching and one-step policies.

---

**Algorithm 1** Dynamic Inference via Horizon Consensus

1: **Input:** horizons $\mathcal{H}$, horizon-wise actions $\{\hat{a}_k\}_{k=1}^H$, MoH fused actions $\hat{a}$, weights $\{\alpha_k\}_{k=1}^H$, scaling ratio $r$, minimum steps $n$, minimum active horizons $m$.
2: **Output:** executable prefix actions $\{\hat{a}_k\}_{k=1}^{K_{\text{exec}}}$.
3: **for** $k = 1$ to $H$ **do**
4: $\quad \mathcal{H}_k \leftarrow \{h \in \mathcal{H} : k \leq h\}$ {active horizons}
5: $\quad \bar{d}_k \leftarrow \sum_{h \in \mathcal{H}_k} \alpha_k \cdot \|\hat{a} - \hat{a}_k\|$ {disagreements}
6: **end for**
7: $thres \leftarrow \text{Mean}(\{\bar{d}_k\}_{k=1}^n) \cdot r$ {threshold}
8: $K_{\text{exec}} \leftarrow n$
9: **for** $k = n + 1$ to $H$ **do**
10: $\quad$ **if** $|\mathcal{H}_k| < m$ **or** $\bar{d}_k > thres$ **then**
11: $\quad\quad$ **break**
12: $\quad$ **end if**
13: $\quad K_{\text{exec}} \leftarrow k$
14: **end for**
15: **return** $\{\hat{a}_k\}_{k=1}^{K_{\text{exec}}}$

---

**Balance loss for horizon utilization.** Without regularization, the gating network may collapse to some preferred horizons, preventing others from contributing. We encourage balanced utilization of horizons in the spirit of load-balancing losses used in the mixture-of-experts domain (Rau, 2019; Fedus et al., 2022).

Let $\alpha_{b,k,h}$ be the gate weight for sample $b$, step $k$, and horizon $h$. Because the set of valid horizons depends on $k$, we partition the temporal dimension by the ordered boundaries $\{0, h_1, \ldots, h_N\}$. For each interval $(h_{i-1}, h_i]$, the active horizons are $\mathcal{H}_i = \{h \in \mathcal{H} : h > h_{i-1}\}$, and $S_i$ denotes the set of steps in this interval. We define the average usage of horizon $h \in \mathcal{H}_i$ as

$$\bar{\alpha}_h^{(i)} = \frac{1}{B |S_i|} \sum_{b=1}^B \sum_{k \in S_i} \alpha_{b,k,h}, \quad (14)$$

where $B$ is the batch size. The balance loss is the mean squared coefficient of variation of these averages:

$$L_{\text{bal}} = \frac{1}{|\mathcal{I}|} \sum_{i \in \mathcal{I}} \text{CV}^2\big(\{\bar{\alpha}_h^{(i)}\}_{h \in \mathcal{H}_i}\big), \quad (15)$$

$$\text{CV}^2(p) = \text{Var}(p) / (\text{Mean}(p)^2 + \varepsilon). \quad (16)$$

where $\mathcal{I}$ indexes intervals with $|\mathcal{H}_i| > 1$ and $\varepsilon$ is a small constant. Minimizing $L_{\text{bal}}$ discourages degenerate gates and ensures that all horizons are effectively utilized.

**Training objective.** Let $L_{\text{mix}}$ denote the loss computed on the fused predictions $\{\hat{a}_{t,k}\}$, and $L_{\text{ind}} = \sum_{h \in \mathcal{H}} L^{(h)}$ the sum of losses on the individual horizon-specific predictions $\{\hat{A}_t^{(h)}\}$. For flow-matching policies, $L_{\text{mix}}$ and $L^{(h)}$ are velocity-matching losses. For one-step policies, they are the corresponding classification or regression objectives introduced in Section 3.1.

*Table 1.* Comparison of VLA models on LIBERO. *Iters* is the abbreviation of training iterations. Best results are in **bold**. Improvements obtained by MoH over respective baselines are marked in (↑). MoH consistently improves flow-matching and regression-based baselines. † UniVLA and X-VLA use large training batch size of 192 and 128, separately.

| Method | Size | Iters | Spatial | Object | Goal | Long | Average |
|---|---|---|---|---|---|---|---|
| *Regression or classification-based VLA* | | | | | | | |
| Octo (Team et al., 2024) | 0.1B | - | 78.9 | 85.7 | 84.6 | 51.1 | 75.1 |
| OpenVLA (Kim et al., 2024) | 7B | 150k | 84.7 | 88.4 | 79.2 | 53.7 | 76.5 |
| CoT-VLA (Zhao et al., 2025) | 7B | 100k | 87.5 | 91.6 | 87.6 | 69.0 | 83.9 |
| $\pi_0$-FAST (Pertsch et al., 2025) | 3B | 30k | 96.4 | 96.8 | 88.6 | 60.2 | 85.5 |
| UniVLA (Bu et al., 2025b) | 9B | 8k† | 96.5 | 96.8 | 95.6 | **92.0** | 95.2 |
| $\pi_{\text{reg}}$ (Black et al., 2024) | 3B | 30k | 97.8 | 98.2 | 94.6 | 90.2 | 95.2 |
| $\pi_{\text{reg}}$ with MoH (Ours) | 3B | 30k | **99.0** ↑1.2 | **98.8** ↑0.6 | **96.4** ↑1.8 | 91.4 ↑1.2 | **96.4** ↑1.2 |
| *Flow-matching or diffusion-based VLA* | | | | | | | |
| Diffusion Policy (Chi et al., 2023) | 30M | - | 78.3 | 92.5 | 68.3 | 50.5 | 72.4 |
| SmolVLA (Shukor et al., 2025) | 2B | 100k | 93.0 | 94.0 | 91.0 | 77.0 | 88.8 |
| GR00T-N1 (Bjorck et al., 2025) | 3B | 100k | 94.4 | 97.6 | 93.0 | 90.6 | 93.9 |
| OpenVLA-OFT(Kim et al., 2025) | 7B | 150k | 97.6 | 98.4 | 97.9 | 94.5 | 97.1 |
| VLA-Adapter(Wang et al., 2025a) | 0.5B | 150k | 97.8 | 99.2 | 97.2 | 95.0 | 97.3 |
| X-VLA (Zheng et al., 2025b) | 1B | 60k† | 98.2 | 98.6 | 97.8 | 97.6 | 98.1 |
| Spatial Forcing (Li et al., 2025a) | 7B | 150k | **99.4** | 99.6 | **98.8** | 96.0 | 98.5 |
| $\pi_0$ (Black et al., 2024) | 3B | 30k | 97.4 | 98.2 | 95.4 | 84.2 | 93.8 |
| $\pi_0$ with MoH (Ours) | 3B | 30k | 97.6 ↑0.2 | 98.8 ↑0.6 | 96.4 ↑1.0 | 87.4 ↑3.2 | 95.1 ↑1.3 |
| StarVLA (StarVLA-Community, 2026) | 3B | 30k | 98.0 | 98.2 | 95.8 | 91.4 | 95.9 |
| StarVLA with MoH (Ours) | 3B | 30k | 98.4 ↑0.4 | 99.6 ↑1.4 | 97.6 ↑1.8 | 92.4 ↑1.0 | 97.0 ↑1.1 |
| $\pi_{0.5}$ (Shi et al., 2025) | 3B | 30k | 98.8 | 99.0 | 97.6 | 95.4 | 97.7 |
| $\pi_{0.5}$ with MoH (Ours) | 3B | 30k | 98.8 ↑0 | **100** ↑1.0 | **98.8** ↑1.2 | **98.4** ↑3.0 | **99.0** ↑1.3 |

The final training objective is

$$L = L_{\text{mix}} + \lambda_{\text{ind}} L_{\text{ind}} + \lambda_{\text{bal}} L_{\text{bal}}, \qquad (17)$$

where $\lambda_{\text{ind}}$ and $\lambda_{\text{bal}}$ are empirically set to 1 and $10^{-3}$.

### 3.3. Dynamic Inference via Horizon Consensus

**Motivation.** Standard inference commits a fixed prefix length from every predicted chunk, regardless of how confident the policy is at the current step. Intuitively, it is better to commit more steps and reduce replanning when the motion is simple and low-risk, and commit fewer steps and replan more often when nearing decision points or during fine-grained manipulation. Fortunately, MoH naturally provides such a signal through horizon-wise prediction agreement.

**Cross-horizon consensus.** We design a dynamic inference scheme via *cross-horizon consensus* for stable and fast inference. As illustrated in Algorithm 1, at each step, horizon-wise predictions $\{\hat{a}_k\}_{k=1}^H$ serve as voters on the fused actions $\hat{a}$. We measure the $\ell_1$ disagreement between $\hat{a}$ and all valid $\hat{a}_k$ with gating weights $\alpha$. The first $n$ steps provide a data-dependent threshold, and we execute the longest action prefix whose disagreement remains below this threshold while enough horizons are still active. This yields a self-truncating executable chunk where only cross-horizon-consistent actions are committed, and the remaining tail is deferred to replanning.

## 4. Simulation Experiments

### 4.1. Experimental Setup

**Simulation Setup.** We evaluate our method on three widely used simulation benchmarks: LIBERO (Liu et al., 2023), RoboTwin2.0 (Chen et al., 2025), and RoboCasa (Nasiriany et al., 2026). LIBERO contains four task suites: Spatial, Object, Goal, and Long. Each suite contains 10 tasks and 500 demonstrations in total, and is designed to probe generalization to different spatial layouts, objects, goals, or long-horizon tasks. RoboTwin is a bimanual benchmark covering 50 diverse tasks. For each task, RoboTwin provides an easy setting with in-domain layouts and a hard setting with domain randomization, including scene clutter, diverse background textures, lighting variations, and different tabletop heights. Due to computational limitations, we evaluate methods on 7 representative tasks from RoboTwin. RoboCasa is a less saturated benchmark designed for generalist robot manipulation in diverse household scenes; we select one task from each of five task suites: *BottleToCabinetClose*, *CuttingboardToBasket*, *PlacematToBasket*, *PlateToBowl*, and *TrayToCardboardbox*. For all benchmarks, we report the success rate as evaluation metric.

**Base Models and Implementation Details.** We select $\pi$ series (Black et al., 2024; Shi et al., 2025) and StarVLA (StarVLA-Community, 2026) as our base models, including flow-matching-based $\pi_0$, $\pi_{0.5}$, StarVLA-GR00T

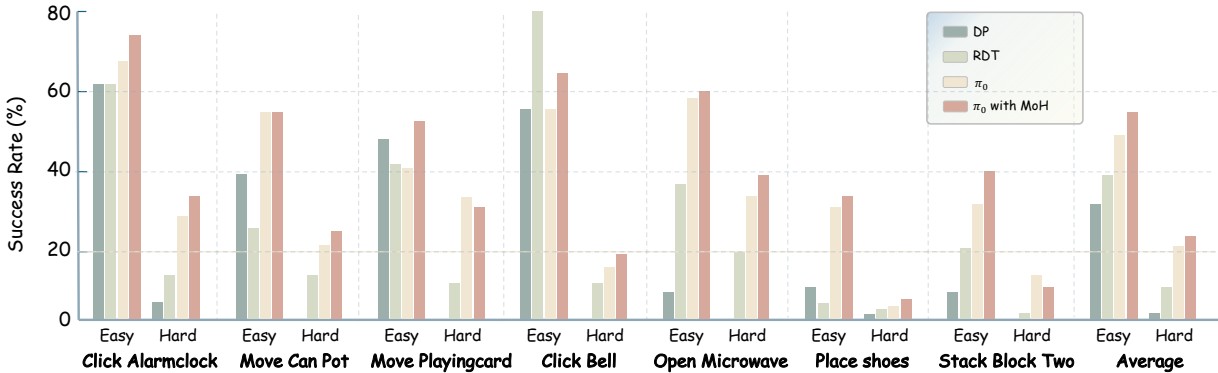

*Figure 4.* Comparisons with state-of-the-art methods on RoboTwin 2.0 Benchmark.

and regression-based $\pi_{\text{reg}}$. Models in $\pi$ series are built upon PaliGemma (Beyer et al., 2024) and pre-trained on large-scale embodied datasets (O'Neill et al., 2024). Since $\pi_0$ does not release a regression-type base model, we obtain $\pi_{\text{reg}}$ by fine-tuning the released $\pi_0$ base model with a regression objective. Its architecture is presented in Appendix D. StarVLA is built on Qwen3-vl (Bai et al., 2025a).

Following the official settings of the $\pi$-series, we train all models once on the mixed LIBERO training set containing all four task suites. Training is performed on 4 NVIDIA A100 GPUs for only $30k$ iterations with a batch size of 32 and a fixed random seed for all comparisons. No historical information (past observations or actions) is provided to the VLA models. Unless otherwise specified, the default horizon configuration for MoH is $\mathcal{H} = \{3, 6, \ldots, 30\}$ with a stride of 3. Each training run finishes in less than 10 hours.

Regarding RoboTwin, again following the official configuration, we train each model for 20 epochs using 50 clean demonstrations per task, training a separate policy for each of the selected tasks. This corresponds to roughly $3k$–$10k$ iterations depending on the task, after which we evaluate the policies on both the easy and hard modes.

For RoboCasa, we train StarVLA-GR00T on 1,000 trajectories (200 demonstrations per task) in total and execute the first 10 steps of each predicted chunk at evaluation, reporting the average success rate and standard deviation across three random seeds with 150 rollouts per task.

### 4.2. Comparisons with Related Work

**LIBERO.** For a fair comparison, we evaluate each task suite with 500 trials, using identical random seeds across all policies. Following the official LIBERO setting, only the first 5 action steps of each predicted chunk are executed during evaluation. As shown in Table 1, our MoH strategy brings consistent and substantial gains to all four baselines. In particular, $\pi_{0.5}$ with MoH attains an average success rate of 99%, establishing a new state of the art on this benchmark.

*Table 2.* Comparisons on RoboCasa (Nasiriany et al., 2026) based on GR00T. We report success rate (%) and standard deviation across three random seeds (150 rollouts per task in total).

| Model | Bottle Cabinet | Cutting-board | Placemat Basket | Plate Bowl | Tray Cardbox | Avg |
|---|---|---|---|---|---|---|
| GR00T | $56.0_{\pm5.3}$ | $20.0_{\pm5.3}$ | $20.7_{\pm5.0}$ | $11.3_{\pm4.2}$ | $32.0_{\pm2.0}$ | $28.0_{\pm1.2}$ |
| + MoH | $\mathbf{60.7_{\pm6.4}}$ | $\mathbf{22.7_{\pm2.3}}$ | $\mathbf{23.3_{\pm7.6}}$ | $\mathbf{16.7_{\pm4.2}}$ | $\mathbf{33.3_{\pm7.6}}$ | $\mathbf{31.4_{\pm1.2}}$ |

These results demonstrate that MoH effectively integrates precise short-horizon control and long-horizon foresight, thereby mitigating the inherent trade-off between them and further boosting overall performance.

We also observe that the regression-based policy $\pi_{\text{reg}}$, obtained by fine-tuning from the $\pi_0$ base model, can even outperform the standard fine-tuned flow-matching-based $\pi_0$. Given that LIBERO's training and evaluation settings are highly in-distribution, this indicates that the regression objective converges well on small-scale downstream tasks, further supporting the soundness of the $\pi_{\text{reg}}$ design.

**RoboTwin.** We evaluate each easy and hard task with 100 trials, using identical random seeds across all policies for a fair comparison. To accelerate evaluation, we execute the prefix 20 action steps of each predicted chunk. As shown in Figure 4, $\pi_0$ equipped with MoH achieves the highest average success rate and consistently improves over the base $\pi_0$ on most tasks, verifying the general effectiveness of our MoH strategy across diverse downstream scenarios. Since models are only trained with easy demonstrations, the gains on both easy and hard variants indicate that MoH not only accelerates in-distribution convergence, but also enhances robustness and generalization to more challenging task configurations.

**RoboCasa.** As shown in Table 2, GR00T with MoH consistently improves over the baseline on all five tasks, yielding an average gain of $3.4\%$. These results demonstrate that MoH's benefits are not specific to near-saturated settings: our strategy continues to generalize to harder, less saturated environments and to distinct VLA backbones.

## 4.3. Ablation Study

We aim to answer five questions in this subsection:

1. How does the horizon density of MoH influence the VLA models' performance?
2. Do MoH's gains arise from horizon diversity or from a generic ensemble effect?
3. Does the loss-reweighting strategy help alleviate the horizon trade-off?
4. How does a simple mean fusion of horizons without a gating network perform?
5. How essential is the gating balance loss, and how does it affect the learned gating weights?

*Table 3.* Ablation on horizon density for MoH with $\pi_{0.5}$ backbone on LIBERO. We fix the maximum horizon $H_{\max} = 30$ and instantiate the candidate set $\mathcal{H} = \{d, 2d, \ldots, H_{\max}\}$ with varying stride $d$. Smaller $d$ corresponds to denser multi-scale horizons.

| Config | $\mathcal{H}$ | Spatial | Object | Goal | Long | Avg |
|---|---|---|---|---|---|---|
| $\pi_{0.5}$ baseline | {30} | 98.8 | 99.0 | 97.6 | 95.4 | 97.7 |
| +MoH $d$=10 | {10,20,30} | 98.8 | 99.8 | 97.6 | 96.8 | 98.3 |
| +MoH $d$=5 | {5,10,...,30} | **99.6** | 99.0 | 98.4 | 96.2 | 98.3 |
| +MoH $d$=3 | {3,6,...,30} | 98.8 | **100** | **98.8** | **98.4** | **99.0** |
| +MoH $d$=2 | {2,4,...,30} | 99.2 | 98.6 | 98.4 | 97.0 | 98.3 |
| +MoH $d$=1 | {1,2,...,30} | 99.0 | 99.4 | 98.4 | 96.2 | 98.3 |

**Effect of horizon density.** To understand how many horizons are needed for effective multi-horizon fusion, we fix $H_{\max} = 30$ and vary the stride $d$ used to construct the candidate set $\mathcal{H} = \{d, 2d, \ldots, H_{\max}\}$. Table 3 compares the single-horizon $\pi_{0.5}$ ($\mathcal{H} = 30$) with MoH variants that use increasingly dense horizon sets. Introducing just three horizons ($d = 10$) already improves the average success rate from 97.7% to 98.3%, indicating that combining a few coarse scales helps reconcile short-term accuracy and long-horizon foresight. Further densifying the candidate horizons leads to consistent gains, and the configuration with stride $d = 3$ achieves the best overall performance (99.0% success), with particularly notable improvements on long-horizon tasks. We also observe that even when the number of horizon groups increases to 15 or 30, the MoH strategy consistently improves over the baseline without causing any training collapse. $\pi_{0.5}$+MoH with stride $d = 3$ provides the strongest overall results, suggesting that more horizon groups are not always better. Instead, choosing an appropriate stride can simultaneously yield strong performance while controlling training and inference cost. Overall, these results show that MoH reliably benefits from access to multiple horizons, and that a moderately dense set of horizons is sufficient to capture complementary temporal structures.

Together with the failure and challenge analyses in Appendix I (Figures 12), these results further suggest that $\pi_{0.5}$ with MoH already achieves sufficiently high success rates on LIBERO. Many of the remaining failures are

*Table 4.* Ablation of mixture and balance strategies for $\pi_{0.5}$ on LIBERO with $H_{\max} = 30$ and stride $d = 3$. Starting from the single-horizon $\pi_{0.5}$ baseline, we compare: (i) MoH with 10 identical horizons (all $H$=30) to isolate the ensemble effect, (ii) temporal loss reweighting without MoH, (iii) uniform mean fusion over valid horizons (no gating head), (iv) MoH without the balance loss $L_{\mathrm{bal}}$, and (v) our full MoH with gated fusion and $L_{\mathrm{bal}}$.

| Variant | Spatial | Object | Goal | Long | Avg |
|---|---|---|---|---|---|
| $\pi_{0.5}$ baseline | 98.8 | 99.0 | 97.6 | 95.4 | 97.7 |
| +MoH w/ identical horizons | 98.6 | 99.4 | 98.6 | 94.8 | 97.9 |
| +Loss reweighting, no MoH | **99.2** | 99.6 | **99.2** | 94.4 | 98.1 |
| +MoH with average fusion | 98.8 | 99.2 | 98.6 | 96.8 | 98.4 |
| +MoH without $L_{\mathrm{bal}}$ | 98.2 | **100** | 99.0 | 96.8 | 98.5 |
| +MoH (ours) | 98.8 | **100** | 98.8 | **98.4** | **99.0** |

largely attributable to environmental issues or limitations in instruction-following, which are outside the scope of what MoH is designed to address.

**Effect of horizon diversity.** To verify that the gains of MoH originate from horizon diversity rather than a generic ensemble effect, we replace the heterogeneous horizon set with 10 identical branches all using $H$=30, keeping the same gating and balance loss. As shown in the second row of Table 4, this configuration yields only a marginal improvement over the single-horizon baseline (97.7% → 97.9%) and crucially fails to alleviate the horizon trade-off on the Long suite. In contrast, MoH with diverse horizons further lifts the average success rate to 99.0% with consistent gains across all suites, confirming that the benefit of MoH stems primarily from fusing predictions across *distinct* horizons, not from simply ensembling multiple parallel branches.

**Effect of loss reweighting.** Table 4 disentangles the contributions of the key MoH components. First, we test whether the gains can be attributed purely to loss weighting rather than multi-horizon modeling. Motivated by the implicit emphasis on early steps induced by the MoH objective, we construct a *loss reweighting only* variant that applies analogous temporal weights directly to the single-horizon $\pi_{0.5}$, without introducing additional horizons or gating. This variant indeed improves performance on three short-term suites, but further degrades the Long suite, thereby intensifying the trade-off. The higher average success rate comes at the cost of long-horizon robustness, confirming that MoH's improvements are not explained by loss reweighting alone.

**MoH with simple average fusion.** Second, replacing gated fusion with naive average fusion over all valid horizons, corresponding to the third line in Table 4, successfully alleviates the short-vs-long-horizon trade-off and yields modest overall improvements over the baseline. This result strongly supports our motivation: even the simplest implementation of MoH already works well, and the additional gating network is expected to further improve performance.

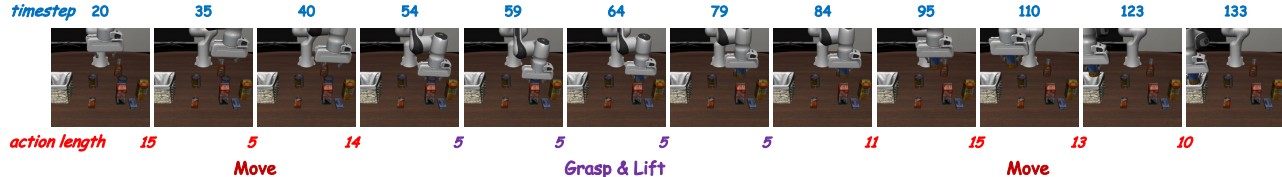

*Figure 5.* Example of dynamic inference on LIBERO-Long. $\pi_{0.5}$ with MoH runs dynamic inference with scaling ratio $r = 1.1$. After each action chunk prediction, only the prefix actions with horizon consensus are executed. Shorter chunks are selected near decision points and fine-grained manipulation, whereas longer chunks are used during smooth, low-risk motions.

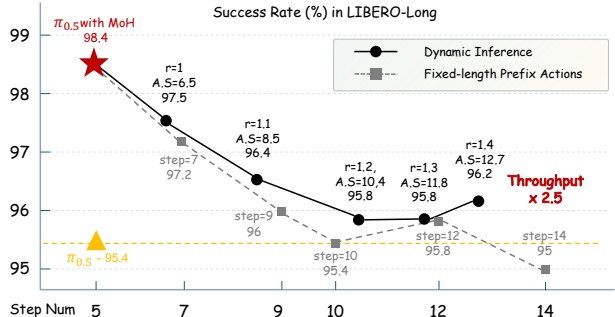

*Figure 6.* Dynamic inference v.s. fixed-length prefix. $A.S$ is abbreviation of average action step number.

**Effect of gating balance loss.** Finally, we evaluate *MoH w/o $L_{bal}$* to assess the role of the balance loss. As shown in line 4 of Table 4, gating without $L_{bal}$ already provides a clear gain over the $\pi_{0.5}$ baseline, showing that learned gated fusion across horizons is inherently beneficial. We also present and analyze the statistics of gating weights at each valid action step in Appendix F.3.

Together, these ablations demonstrate that (i) horizon diversity, (ii) learnable gated fusion, and (iii) gating balance regularization jointly contribute to robustly alleviating the horizon trade-off problem.

### 4.4. Effect of Dynamic Inference

We compare the dynamic inference scheme introduced in subsection 3.3 with using a fixed-length prefix. We define *throughput* as the average number of action steps executed per predicted chunk, i.e., the utilization rate of the generated actions rather than wall-clock test time or per-forward inference latency, and use it as the efficiency metric throughout this subsection. By default, we set $n = 5, m = 5, d = 3$ and then change the value of $r$ to observe the corresponding performance of $\pi_{0.5}$ with MoH on LIBERO-Long. See Figure 6, dynamic inference consistently outperforms the basic fixed-length strategy. Notably, even when throughput is increased to 2.5× the default setting (5 steps), $\pi_{0.5}$ with MoH under dynamic inference still outperforms the baseline $\pi_{0.5}$. As illustrated in Figure 5, dynamic inference enables robots to move quickly when motion is simple and risk is low, while acting more cautiously and updating more frequently during critical decision-making and precise manipulation.

*Table 5.* Comparison with Adaptive Temporal Ensemble (ATE) on LIBERO. ATE yields only marginal gains on short-term suites and consistently degrades long-horizon performance, while MoH improves all four suites robustly.

| Variant | Spatial | Object | Goal | Long | Avg |
|---|---|---|---|---|---|
| $\pi_0$ | 97.4 | 98.2 | 95.4 | 84.2 | 93.8 |
| + ATE | 97.6 | 98.2 | 94.8 | 83.4 | 93.5 |
| + MoH | 97.6 | 98.8 | 96.4 | **87.4** | **95.1** |
| + MoH & ATE | 97.0 | 98.6 | 96.6 | 86.6 | 94.7 |
| $\pi_{0.5}$ | 98.8 | 99.0 | 97.6 | 95.4 | 97.7 |
| + ATE | 98.6 | 99.4 | 97.8 | 90.6 | 96.6 |
| + MoH | 98.8 | **100** | **98.8** | **98.4** | **99.0** |
| + MoH & ATE | 98.8 | 99.6 | 98.6 | 96.4 | 98.4 |

### 4.5. Comparison with Temporal Ensemble Strategy

Action temporal ensembling (Zhao et al., 2023; Li et al., 2024) smooths execution by fusing predictions of the same timestep from successive chunks. Since MoH also operates by fusing action sequences, we compare against CogACT's *Adaptive Temporal Ensemble* (ATE) (Li et al., 2024), a widely used similarity-weighted variant, to disentangle whether MoH's gains stem from a similar smoothing effect or from a deeper improvement in predictive capability. Conceptually, MoH and ATE are orthogonal: MoH fuses across *different horizons within a single planning step*, whereas ATE fuses across *actions at the same timestep but from different decision steps*. The two are therefore not in conflict and can be applied jointly, as shown by the "MoH & ATE" rows in Table 5.

As shown in Table 5, ATE alone yields only marginal gains on short-term suites and consistently degrades long-horizon performance, most notably dropping $\pi_{0.5}$ on Long from 95.4 to 90.6. We attribute this to a cumulative *drift* effect: smoothing helps a noisy policy, but for a precise one it introduces an offset relative to the current optimal plan that compounds over long horizons. MoH, in contrast, improves all four suites robustly on both backbones, indicating that it enhances the underlying temporal modeling rather than merely smoothing outputs. Stacking ATE on top of MoH brings no additional benefit and slightly hurts the performance on Long suite, since once MoH already produces well-aligned predictions, ATE's smoothing mostly reintroduces drift without compensating gains.

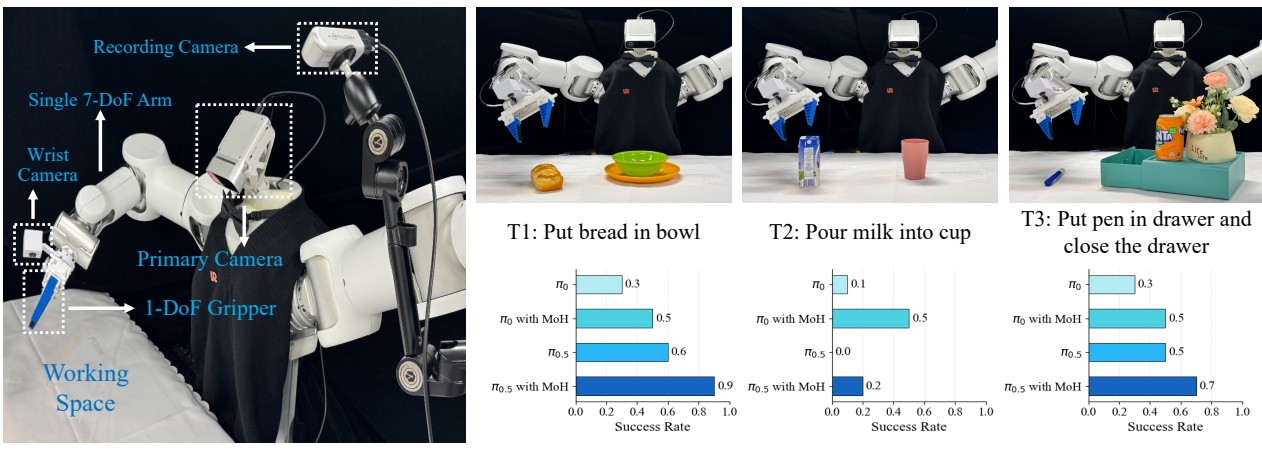

(a) Platform Setup

(b) Task Setting and Corresponding Results

*Figure 7.* Experimental settings and results in real-world scenarios.

## 5. Real-world Experiments

**Platform and Task Setup.**

We conduct real-robot experiments on a self-developed platform to evaluate the effectiveness of MoH. As shown in Figure 7 (a), we adopt the single arm setting, which consists of a 7-DoF manipulator and 1-DoF gripper. A primary camera and a wrist camera are installed to provide visual observations for VLA models, and a third-view camera is used to record task completion process. We design four evaluation tasks: two short-horizon tasks, *T1: put bread into the bowl* and *T2: pour milk into the cup*; one long-horizon task, *T3: put the pen into the drawer and close it*; and one fine-grained deformable-manipulation task, *T4: fold the towel diagonally*. Together they require instruction following, object relocation and rotation, precise grasping/placement, and deformable manipulation, providing a comprehensive evaluation of VLA models in real-world settings.

**Base Models and Implementation Details.** We adopt $\pi_0$ and $\pi_{0.5}$ as base models. For each of T1–T3 we collect 30 expert demonstrations, and for T4 we collect 40 teleoperated demonstrations. All models are trained for $10k$ iterations with a batch size of 32, executing the prefix 5 actions of each predicted chunk by default. To ensure fair evaluation, models with and without MoH are executed sequentially from the same initial scene configurations. After each pair of rollouts, we perturb object poses, goal locations, and orientations. T1–T3 are evaluated with 10 rollouts per task. The limitations of action steps are set to 2000 steps for short-horizon tasks and 3000 steps for the long-horizon task. T4 are evaluated on $\pi_{0.5}$ for 20 rollouts.

**Result and Analysis.** As shown in Figure 7(b), MoH yields consistent gains across all tasks and both base models, improving both long-horizon decision making and short-horizon fine-grained action prediction. On T1, baseline

policies typically exhibit back-and-forth hesitation before committing to a grasp, while MoH approaches the bread directly and executes a faster, more decisive motion. On the long-horizon T3, MoH also leads to more accurate grasps, which in turn yields quicker completion and higher success rates. On the fine-grained T4, $\pi_{0.5}$ improves from $75\%$ (15/20) to $90\%$ (18/20) with MoH, confirming that horizon mixture particularly benefits tasks requiring phase-sensitive precise control such as folding a deformable object.

We also notice an interesting phenomenon on the *pour milk into cup* task: $\pi_{0.5}$ performs worse than $\pi_0$. A closer inspection reveals that, after lifting the milk bottle, $\pi_{0.5}$ often hesitates between continuing to pour and putting the bottle back. Since both actions appear in the training set and the policy does not receive explicit action history as input, this suggests that $\pi_{0.5}$ overfits this local conflict during training. In contrast, equipping the model with MoH helps alleviate this overfitting, enabling a clearer modeling of short-range motions and long-horizon intent.

Overall, the real-world experiments are consistent with our simulation results, confirming that MoH effectively combines long-horizon planning with short-term control for practical robotic manipulation.

## 6. Conclusion

In this paper, we introduced Mixture of Horizons, a plug-and-play strategy that fuses multi-horizon action chunks in full-attention VLA policies to ease the trade-off between long-term foresight and short-term precision. Across simulator benchmarks and real-world tasks, MoH consistently improves both flow-matching and one-step regression policies, achieving a new state of the art on LIBERO with $\pi_{0.5}$. Ablations confirm the benefits of dense horizons, gated fusion, gating balance regularization and dynamic inference.

## Impact Statement

This paper provides a systematic analysis on a long-overlooked but critical factor of VLAs: a single action chunk horizon induces a trade-off between long-term foresight and short-term control fidelity. The proposed Mixture of Horizons (MoH) is a plug-and-play and computationally efficient framework that consistently alleviates such trade-offs and enables a more flexible dynamic inference for adaptive closed-loop control. We emphasize and demonstrate that the simple yet effective nature of MoH makes it generalize broadly to any base models and diverse settings. Beyond performance gains, MoH offers a principal pathway towards more robust VLA pretraining hence more scalable foundational action models. Our insights can potentially be applied to agentic AI, world models, and interactive perception systems, benefiting the large embodied and agentic AI communities.

## Acknowledgements

This work is partially supported by National Natural Science Foundation of China (62376274, 62437002).

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

# A. Training hyperparameters

*Table 6.* Training hyperparameters of $\pi$ series on LIBERO.

| Hyperparameter | Value | Hyperparameter | Value |
|---|---|---|---|
| GPUs | $4 \times$ A100 | Optimizer | AdamW |
| Iterations | 30k | Total Batch Size | 32 |
| Learning Rate | 5e-5 | Minimum LR | 1e-6 |
| Scheduler | Warmup & Cosine Decay | Warmup Step | 1k |

In Table 6, we present hyperparamters used to train $\pi_0$, $\pi_{0.5}$ and $\pi_{reg}$ on LIBERO mixed dataset. Regarding the RoboTwin, we only adjust the learning rate to $2.5e-5$ and train models for 20 epochs on 50 clean demos for each task.

# B. Latency Comparison

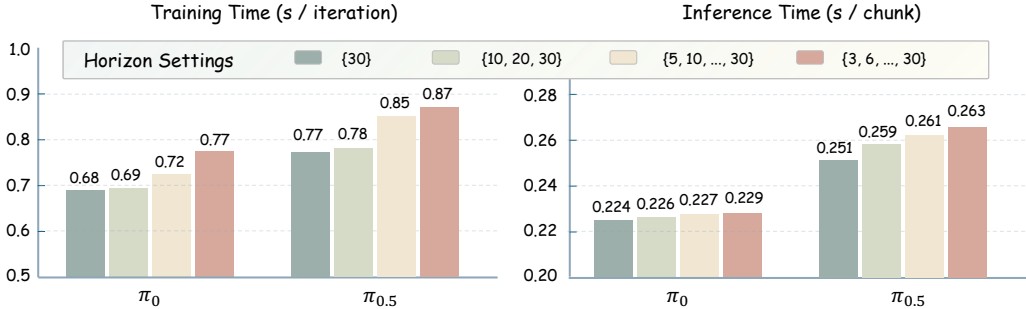

*Figure 8.* Visualization of the overhead under different horizon settings. Since the action transformer is typically lightweight, and combined with tensor parallelism, MoH incurs very little additional overhead for both training and inference.

In Figure 8, we present the training and inference time cost of $\pi_0$ and $\pi_{0.5}$ under different horizon settings. Benefiting from data parallelism, MoH brings very little additional time overhead for both training and inference. Importantly, the inference latency is virtually unaffected, which means that MoH does not impact the control frequency and fully preserves the usability of VLA models.

# C. Discussion of MoH Hyper-parameter Setup in Pre-training and Post-training

While this work primarily validates the effectiveness of MoH in the fine-tuning (post-training) stage, we believe its benefits can be extended naturally to VLA pre-training. Large-scale pre-training datasets typically aggregate demonstrations from hundreds or even thousands of diverse environments. Given such immense variability, identifying a single optimal horizon that generalizes across all tasks and control requirements is inherently difficult.

Our empirical results demonstrate that MoH consistently outperforms single-horizon baselines regardless of the specific configuration. This suggests a flexible, two-tier strategy for deploying MoH across the model's whole training process:

1. Pre-training: To balance performance gains with computational overhead, a sparse horizon configuration (e.g., three horizons) can be adopted. As shown in Figure 8, such a configuration incurs a negligible increase in training overhead—approximately 1.4%, while still providing the foresight and precision benefits missing from single-horizon training.

2. Downstream Post-training: In specific task suites where reaching the performance ceiling is the priority, a denser horizon configuration can be utilized. While this increases training overhead by roughly 10%, it allows the model to achieve superior fine-grained control and long-term planning stability for specific complex manipulations.

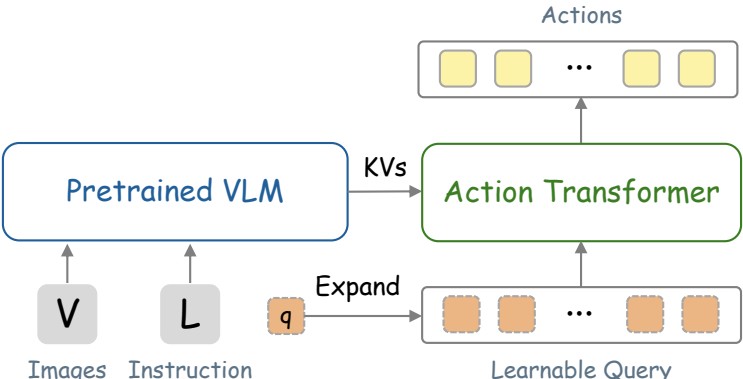

*Figure 9.* Illustration of our designed $\pi_{reg}$, with little modification based on $\pi_0$. We introduce a learnable query token as query input for action transformer. Actions are predicted in one forward pass.

## D. Design of $\pi_{reg}$

Since the $\pi_0$ project does not release its regression type, we obtain the $\pi_{reg}$ by finetuning from the $\pi_0$ base model. As shown in Figure 9, we introduce a learnable query $q$ and expand it to the length of action chunk to serve as the input queries for action transformer. The action chunks are predicted in only one forward process. The training objective isa continuous regression function introduced in Section 3.1.

## E. Example of Dynamic Inference

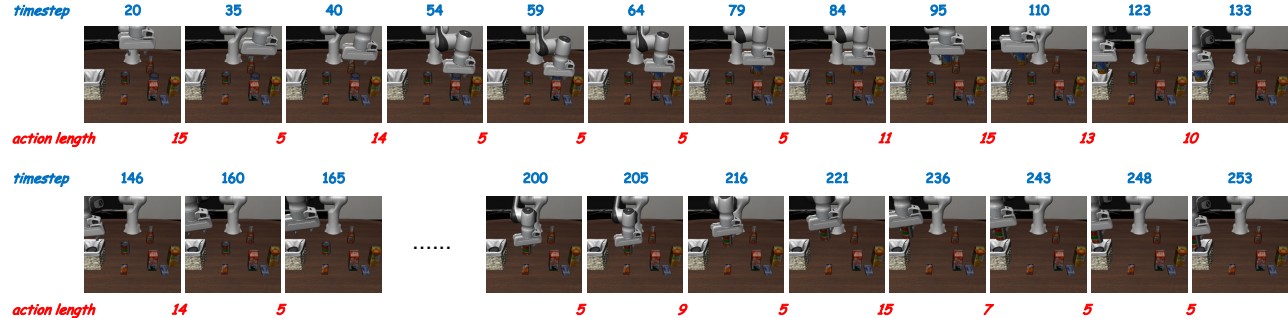

*Figure 10.* Example of dynamic inference on LIBERO-Long. $\pi_{0.5}$ with MoH runs dynamic inference with scaling ratio $r = 1.1$. After each action chunk prediction, only the prefix actions with horizon consensus are executed. Shorter chunks are selected near decision points and fine-grained manipulation, whereas longer chunks are used during smooth, low-risk motions.

In Figure 10, we visualize one rollout on LIBERO-Long under dynamic inference. For this trajectory, we display most timesteps together with the action-chunk lengths that are actually executed. A clear pattern emerges: around decision points, such as when the robot changes its movement direction or commits to approaching a new target object, and during fine-grained manipulation (e.g., grasping and lifting the bottle), the policy tends to select only the shortest horizon of 5 steps. In contrast, when the system is in a relatively stable and low-risk phase, such as translating the grasped object or moving the arm through free space toward a pre-grasp configuration, the executed chunks become noticeably longer. This behavior directly aligns with the motivation behind dynamic inference: allowing the agent to move quickly when the task is simple and risk is low, while acting more cautiously and updating its plan more frequently during critical decision-making and precise manipulation. The qualitative evidence here also suggests that MoH-based dynamic inference implicitly captures task phases and uncertainty, highlighting its potential to balance efficiency and robustness in long-horizon control.

# F. More Ablation Study

## F.1. Trade-off Effect of Single Horizon on $\pi_{0.5}$

*Table 7.* Effect of action horizon on $\pi_{0.5}$. The first 5 actions in the predicted chunk are executed at evaluation. Our mixture of horizons $\in \{10, 20, 30\}$ strategy alleviates the trade-off caused by varying horizons and raises overall success.

| Horizon | Spatial | Object | Goal | 10 | Average |
|---|---|---|---|---|---|
| 10 | **99.0** | 98.8 | **98** | 92.4 | 97.1 |
| 20 | 98.8 | 98.2 | 97.6 | 94.6 | 97.3 |
| 30 | 98.8 | 99.0 | 97.6 | 95.4 | 97.7 |
| MoH | 98.8 | **99.8** | 97.6 | **96.8** | **98.3** |

In the Introduction 1, we analyzed the horizon effect on $\pi_0$. In this section, we provide a further study based on $\pi_{0.5}$, with results shown in Table 7. As the horizon increases, $\pi_{0.5}$ exhibits a similar trade-off across the four tasks: performance on short-horizon tasks fluctuates, while performance on long-horizon tasks steadily improves. In contrast, our MoH strategy effectively mitigates this trade-off and substantially improves the overall success rate.

## F.2. Effect of Maximum Horizon in MoH

*Table 8.* Effect of maximum horizon $H_{max}$ on $\pi_{0.5}$ performance. All MoH variants use 10 horizon groups. Results show that MoH consistently outperforms single-horizon baselines, with $H_{max}$=30 serving as the optimal temporal span for LIBERO task suites.

| Model | Horizon | Spatial | Object | Goal | 10 | Average |
|---|---|---|---|---|---|---|
| $\pi_{0.5}$ | 10 | **99.0** | 98.8 | 98 | 92.4 | 97.1 |
| $\pi_{0.5}$ with MoH | {1,2,...,10} | 98.8 | 99.2 | **98.8** | 93.2 | 97.5 |
| $\pi_{0.5}$ | 30 | 98.8 | 99.0 | 97.6 | 95.4 | 97.7 |
| $\pi_{0.5}$ with MoH | {3,6,...,30} | 98.8 | **100** | **98.8** | **98.4** | **99.0** |
| $\pi_{0.5}$ | 50 | 98.0 | 98.8 | 97.2 | 96.0 | 97.5 |
| $\pi_{0.5}$ with MoH | {5,10,...,50} | 98.4 | 99.6 | 98 | 96.6 | 98.2 |

In Section 4.3, we investigated how horizon density (stride) affects performance. Here, we further explore the impact of the maximum horizon $H_{max}$ within the MoH framework. To ensure a fair comparison, we maintain a constant number of 10 horizon groups while varying $H_{max}$ across $\{10, 30, 50\}$.

As shown in Table 8, MoH provides consistent performance gains over the single-horizon baselines regardless of the $H_{max}$ choice. Notably, the configuration with $H_{max}$=30 achieves the superior results on the LIBERO benchmark. These results suggest that the influence of $H_{max}$ mirrors the fundamental trade-off introduced in Section 1: while a smaller $H_{max}$ (e.g., 10) benefits fine-grained local control, it lacks the necessary global foresight for complex long-horizon tasks. Conversely, an excessively large $H_{max}$(e.g., 50) may introduce planning noise that slightly degrades precision. MoH effectively navigates this trade-off, with a moderate $H_{max}$ providing the balance between short-term accuracy and long-term planning.

## F.3. Statistical Analysis of Gating Balance Loss

We also present the statistics of gating weights at each valid action step on the LIBERO-Long task suite in Figure 11. Without $L_{bal}$, the gate head tends to assign higher weights to action chunks with longer horizons, because longer horizons participate in more steps during action mixture. This introduces statistical and gradient bias during training and manifests as an imbalance in gating learning. After introducing $L_{bal}$, this bias is effectively suppressed, enabling the gating head to better leverage predictions from each horizon. Meanwhile, because $L_{bal}$ acts only as a regularization term, it does not forcibly flatten the weights, thereby avoiding excessive averaging.

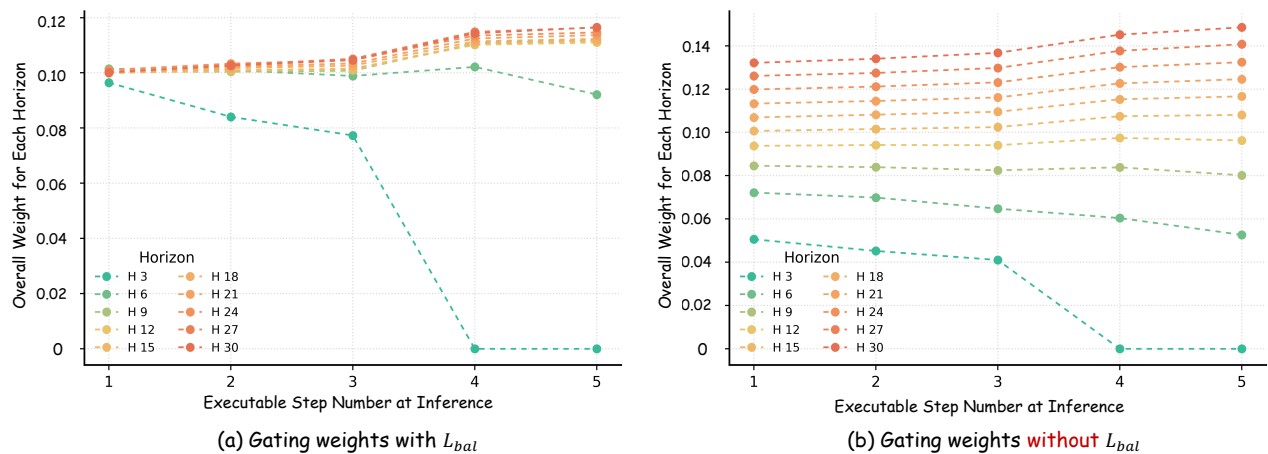

(a) Gating weights with $L_{bal}$  (b) Gating weights without $L_{bal}$

*Figure 11.* Visualization of horizon weights of $\pi_{0.5}$ with MoH on LIBERO-Long task suite. The regulation term $L_{bal}$ encourages the distribution balance across horizons. Without $L_{bal}$, the gating weights present obvious distribution preference at all times. The weights of $H3$ drop to $0$ at steps $4$ and $5$ as it is no longer active.

## G. Empirical Evidence for the Horizon Trade-off

In Section 1, we identified an intrinsic trade-off in action chunking: longer horizons enhance long-term foresight but sacrifice short-term motor precision, and vice versa. This section provides two pieces of direct empirical evidence that (i) the trade-off manifests at the per-step action level (Section G.1), and (ii) MoH learns task-adaptive horizon weights that are consistent with this trade-off (Section G.2).

### G.1. Per-Step Action Error Analysis

*Table 9.* Average $\ell_1$ error on the first 5 actions of each predicted chunk across the LIBERO training set, based on single-horizon $\pi_{0.5}$. The per-step error grows monotonically with the training horizon on every short-term suite (Spatial, Object, Goal), providing direct evidence that longer horizons sacrifice low-level motor precision.

| Horizon | Spatial | Object | Goal | Long | Average |
|---------|---------|--------|--------|--------|---------|
| 10 | 0.0121 | 0.0117 | 0.0129 | 0.0149 | 0.0129 |
| 20 | 0.0122 | 0.0117 | 0.0132 | 0.0149 | 0.0130 |
| 30 | 0.0125 | 0.0118 | 0.0133 | 0.0149 | 0.0131 |

To complement the success-rate based analysis in the main text, we directly measure the $\ell_1$ regression error between predicted and ground-truth actions on the LIBERO training set. For each single-horizon $\pi_{0.5}$ checkpoint (horizon $\in \{10, 20, 30\}$), we compute the per-step $\ell_1$ error over the first 5 actions of each predicted chunk.

As shown in Table 9, the average per-step $\ell_1$ error increases monotonically with the training horizon on every short-horizon suite (Spatial, Object, Goal) and remains essentially flat on the long-horizon Long suite. This offers clean, success-rate-independent evidence supporting our central claim: longer horizons sacrifice low-level motor precision. Moreover, the non-monotonic fluctuations of success rate on short-horizon suites in Figure 1 are consistent with this finding: because these suites are already near saturation, small per-step errors translate into noisy and non-uniform changes in end-task success, making per-step error a cleaner proxy for the underlying trade-off.

### G.2. Task-Adaptive Gating Weights

To further interpret what MoH actually learns, we analyze the gating weights of $\pi_{0.5}$ with MoH across the four LIBERO task suites. For each rollout, we average the gating weights of the first 3 executed actions, representing the model's preferred horizon allocation at decision time.

*Table 10.* Average gating weights (%) over the first 3 executed steps across the four LIBERO task suites for $\pi_{0.5}$ with MoH (each row sums to 100). As task complexity grows from Spatial (shortest trajectories) to Long (longest trajectories), the learned weights gradually shift from the shortest horizon ($h$=3) toward longer horizons (e.g., $h$=30), consistent with the horizon trade-off in Figure 1.

| Suite | $h$=3 | 6 | 9 | 12 | 15 | 18 | 21 | 24 | 27 | 30 |
|-------|-------|------|-------|-------|-------|-------|-------|-------|-------|-------|
| Spatial | 9.09 | 9.96 | 10.09 | 10.09 | 10.07 | 10.08 | 10.09 | 10.14 | 10.18 | 10.21 |
| Object | 8.98 | 9.91 | 10.09 | 10.12 | 10.10 | 10.10 | 10.12 | 10.16 | 10.19 | 10.22 |
| Goal | 8.95 | 9.87 | 10.07 | 10.12 | 10.11 | 10.12 | 10.14 | 10.18 | 10.22 | 10.24 |
| Long | 8.93 | 9.87 | 10.08 | 10.13 | 10.12 | 10.12 | 10.14 | 10.17 | 10.21 | 10.23 |

Table 10 reports the average weight per horizon for each suite, with rows summing to $100\%$. Although the absolute differences are subtle (MoH is regularized by $L_{\text{bal}}$ to maintain balanced utilization), a clear and consistent trend emerges: as task complexity increases from Spatial to Long, the weight on the shortest horizon ($h$=3) drops from 9.09 to 8.93, while the weight on the longest horizon ($h$=30) rises from 10.21 to 10.24. This task-adaptive behavior provides intuitive interpretability for MoH, showing that the gating network learns to reallocate horizon budget toward longer horizons when tasks demand more foresight, in line with the trade-off analysis in Figure 1.

## H. Threshold Design in Dynamic Inference

*Table 11.* An instance of cross-horizon disagreement thresholds along one rollout in the LIBERO Spatial task suite. The threshold varies substantially across different execution phases, motivating a data-dependent rather than fixed global value.

| Num. Step | 5 | 10 | 15 | 29 | 34 | 47 | 52 | 57 | 62 | 73 | 87 |
|-----------|-------|-------|-------|-------|-------|-------|-------|-------|-------|-------|-------|
| Threshold | 0.069 | 0.059 | 0.056 | 0.074 | 0.082 | 0.081 | 0.057 | 0.064 | 0.079 | 0.089 | 0.110 |

In Section 3.3, our dynamic inference scheme uses the mean cross-horizon disagreement of the first $n$ steps of each predicted chunk as a data-dependent threshold for truncation, rather than a single globally fixed value. Here we provide an empirical justification for this design.

As shown in Table 11, the cross-horizon disagreement varies substantially across different execution phases within a single rollout, ranging from 0.056 in well-aligned segments to 0.110 near challenging transitions. This variation reflects the changing difficulty and uncertainty of the underlying manipulation task: smooth motions yield low disagreement, while decision points and fine-grained sub-tasks induce higher disagreement. Consequently, any fixed global threshold would either over-truncate during easy phases (sacrificing throughput) or under-truncate during hard phases (sacrificing safety).

Computing the threshold from the first $n$ steps of each chunk offers three practical advantages. First, it provides a lightweight, data-dependent reference that automatically adapts to the current task and policy state without additional learning. Second, it conveniently fixes the minimum number of executed steps per chunk to $n$, which makes the dynamic scheme directly comparable to fixed-length prefix baselines. Third, by tying the threshold to the current chunk, it remains responsive to phase changes without lagging behind a slow-moving running statistic.

## I. Challenge and Failure Case Analysis

### I.1. LIBERO

We manually inspect rollouts in LIBERO and identify four predominant failure modes, three are illustrated in Figure 12.

The first two stem from artifacts of the environment rather than the VLA policy. In (a), the robot successfully completes the task, but the simulator fails to register success and the episode is terminated when the maximum number of action steps is reached. In (b), a collision bug in the Spatial task suite causes the bowl and ramekin to become stuck together, making the task unsolvable. If we ignore these two environment-induced errors, $\pi_{0.5}$ with MoH would achieve an impressive 99.8% success rate on LIBERO-Spatial, instead of the 98.8% reported in the main table.

Panel (c) illustrates the third type of failure, where the model misidentifies the target object, revealing remaining limitations in the visual perception and instruction-following capabilities of current VLA models. Our MoH strategy is not designed to directly address these perception- and language-understanding issues.

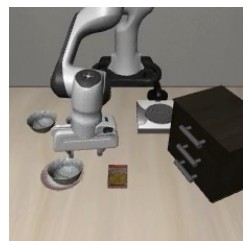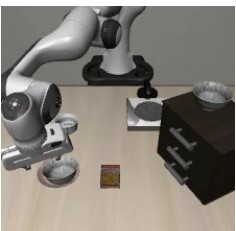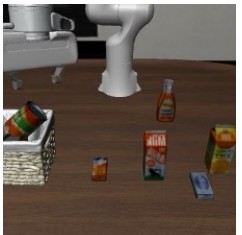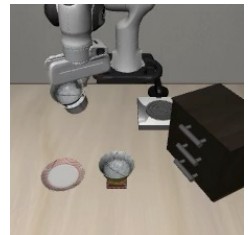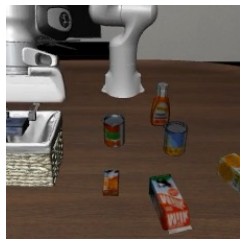

TASK: pick up the black bowl between plate and ramekin and place it on the plate.

TASK: pick up the black bowl next to the plate and place it on the plate.

TASK: put both the alphabet soup and the tomato sauce in basket.

TASK: pick up the black bowl between plate and ramekin and place it on the plate.

TASK: put both the alphabet soup and the tomato sauce in basket.

(a) Task completed but not recognized    (b) Stuck bowl and ramekin    (c) Misidentified target

*Figure 12.* Typical failure modes in LIBERO.

The fourth and most frequent failure mode arises from insufficient low-level action precision, for which we provide demonstrations in the supplementary videos.

## I.2. RoboTwin

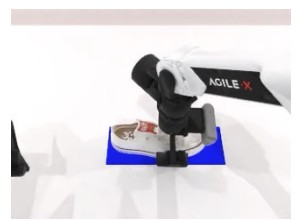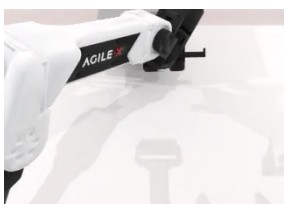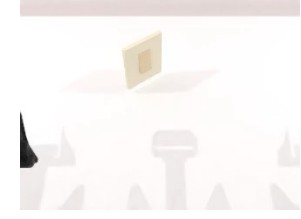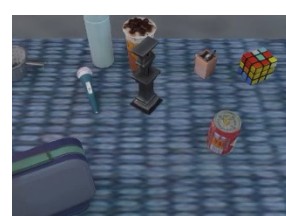

TASK: place shoe

(a) Task completed but not recognized

TASK: open microwave

(b) Heavy Occlusion

TASK: turn switch

(c) Out of observation scope

TASK: click alarm clock

(d) Illegible target

*Figure 13.* Challenges and issues exist in RoboTwin simulator.

Figure 13 highlights several challenging factors and potential issues we observe in RoboTwin2.0 simulator.

First, Figure 13(a) shows that RoboTwin can also fail to signal task completion even when the robot has clearly succeeded. Based on manual inspection of the termination code, we find that some success conditions are overly strict, and small errors in the absolute position thresholds can even bias the states that are labeled as successful.

Figure 13(b) illustrates a scene in `open microwave` task with severe occlusion. In the view, the manipulator completely blocks the target object, making it difficult for the model to correctly infer the current state from the observation. Providing richer history information should help the policy make better decisions and control in such cases, thereby improving success rates in heavily occluded scenes.

In Figure 13(c), many RoboTwin tasks start with the robot arm entirely outside the main camera's field of view. In this situation, VLA models can only infer the arm's pose and state from its shadow or the proprioceptive inputs. We regard this as a rather extreme setting: under normal circumstances, the camera configuration should provide sufficient informative observations. Otherwise, the model may learn shortcuts tailored to this special case, which is undesirable for generalization across diverse scenes.

Under the `random` setting, RoboTwin randomizes the background, object locations, and orientations. Figure 13(d) shows an example of illegible target from the *click alarm clock* task where the alarm clock is placed facing away from the camera. From this viewpoint it is very hard for the model to recognize the object as an alarm clock, and the button region is barely visible, which often leads to failures. This suggests that there exist a non-trivial number of scenes that are extremely difficult for VLA models to solve.

## J. Future Directions

There are two natural directions for extending MoH. *(i) Memory mechanisms.* Incorporating historical observations or a dedicated memory module could improve task-progress tracking and reduce state ambiguity, particularly in long-horizon real-world scenarios. *(ii) Refined dynamic gating.* Enhancing the balance mechanism to enable more adaptive, context-aware weight allocation across horizons, e.g., conditioning the gate on uncertainty or observed task state.

## K. Demonstrations

To better illustrate the behavior of our policy, we present qualitative rollouts produced by $\pi_{0.5}$ with MoH in both simulated and real environments. Figure 14 shows representative executions on several challenge LIBERO tasks and on our real-world setup. The policy is able to predict precise low-level motions and complete long-horizon, multi-stage goals. Figure 15 further visualizes trajectories on RoboTwin 2.0. These qualitative demonstrations show that $\pi_{0.5}$ with MoH is capable to process diverse in-domain tasks and generalized to complex unseen environments.

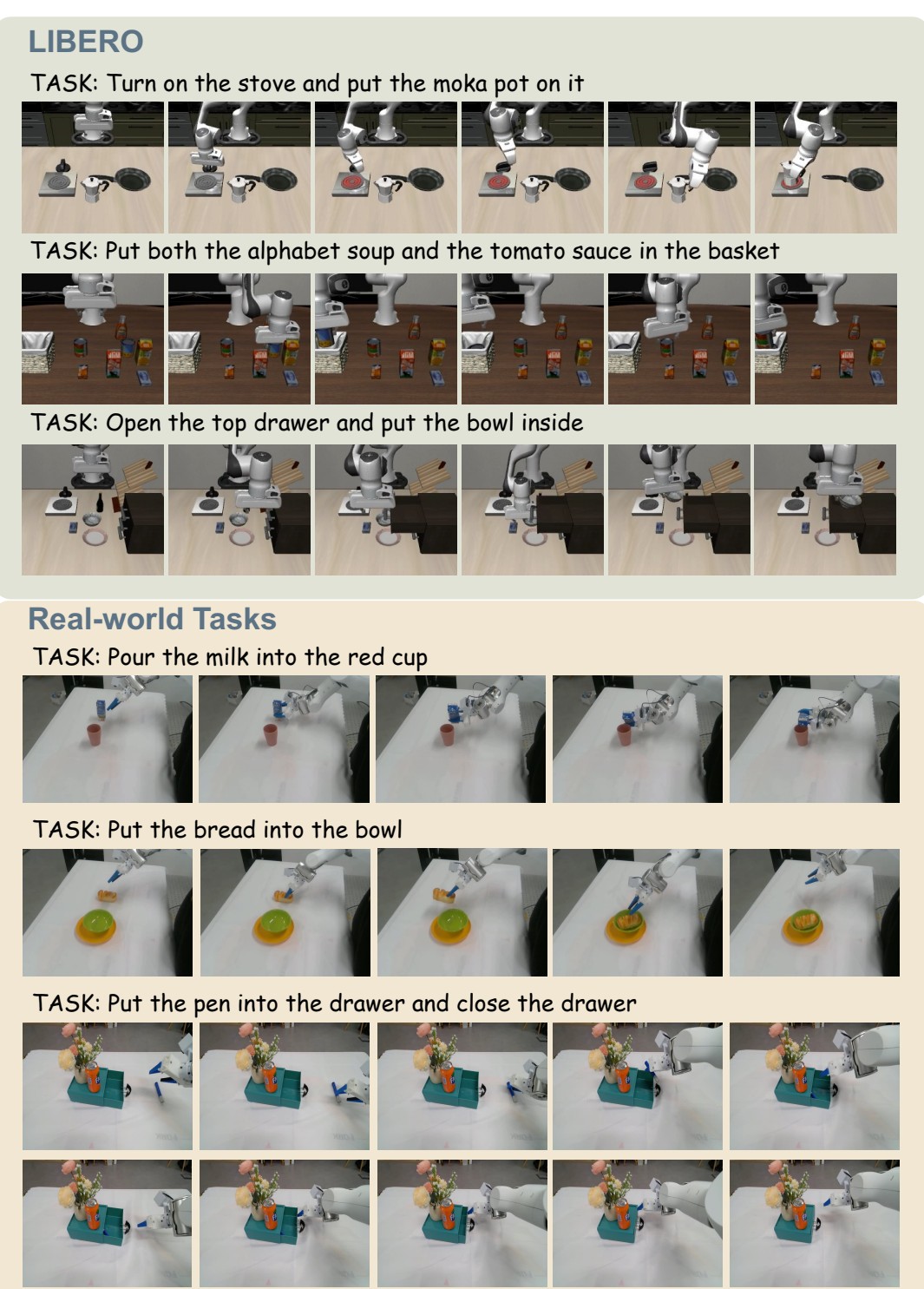

*Figure 14.* Qualitative demonstrations of $\pi_{0.5}$ with MoH on LIBERO and real-world tasks.

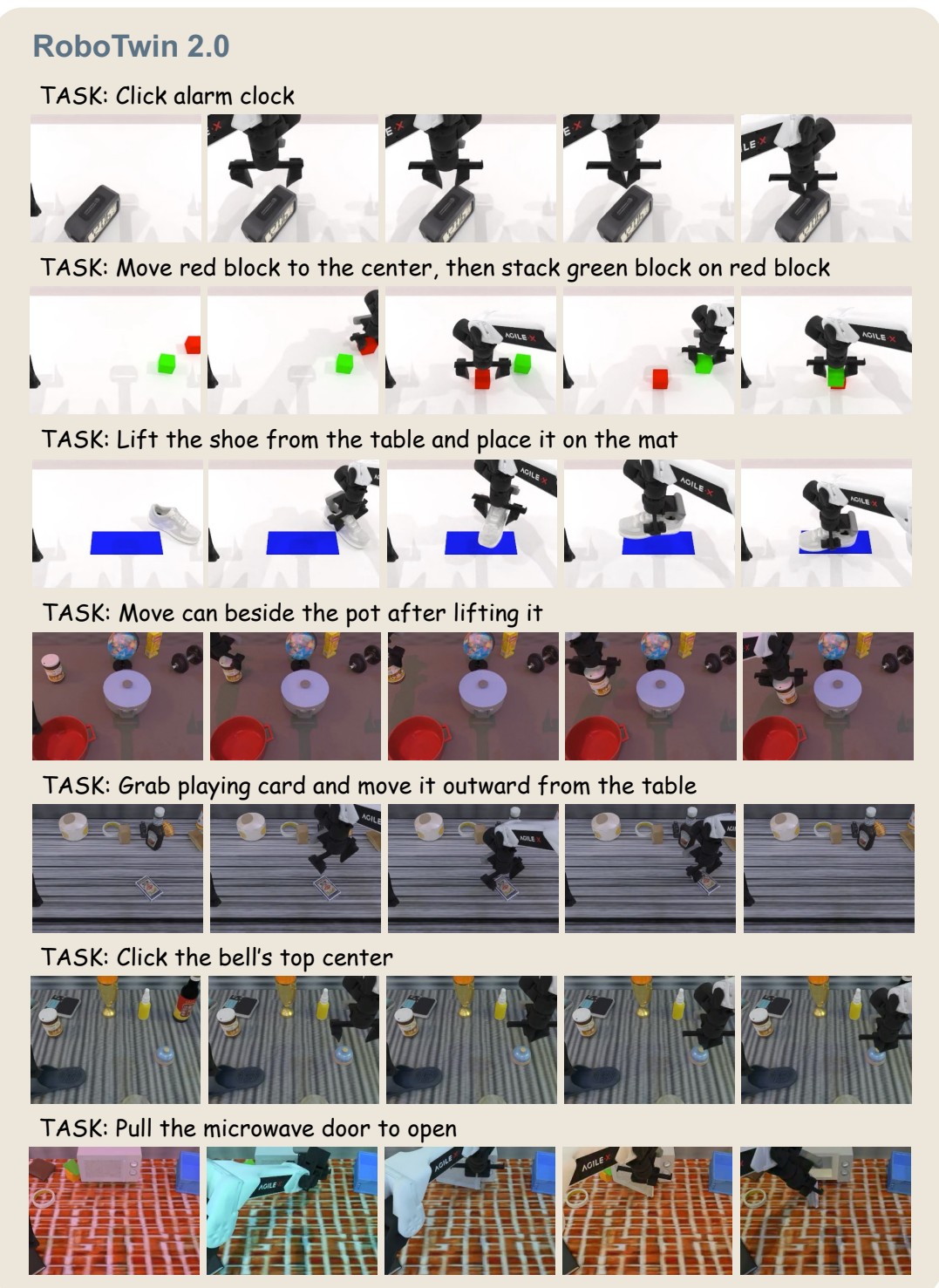

*Figure 15.* Qualitative demonstrations of $\pi_{0.5}$ with MoH on RoboTwin 2.0. The last four lines are collected under `random` settings.

