# OpenReview forum: "Mixture of Horizons in Action Chunking"
_ICML.cc/2026/Conference — ICML 2026 regular_

### Official Review · Reviewer_EmDe · 2026-02-27

**Soundness:** 3
**Presentation:** 3
**Significance:** 3
**Originality:** 3
**Overall Recommendation:** 4
**Confidence:** 4

**Summary:**

This paper proposes Mixture of Horizons (MoH), a plug-and-play method for action chunking in VLA models that alleviates the trade-off between long-term foresight and short-term precision by fusing predictions from multiple action horizons. By introducing horizon-wise parallel processing with a lightweight gating mechanism and a dynamic inference scheme based on cross-horizon consensus, the method consistently improves performance across simulation benchmarks and real-world robotic tasks.

**Compliance With Llm Reviewing Policy:**

Affirmed.

**Key Questions For Authors:**

1. Can the authors provide stronger empirical justification for using the mean disagreement of the first $n$ steps as the threshold?
2. How does MoH perform on more fine-grained manipulation tasks (e.g., cloth folding or deformable object manipulation)? Does it bring larger gains in scenarios that require frequent switching between long-horizon planning and precise short-term control?
3. How does MoH perform on VLA architectures beyond the $\pi$ models?

**Limitations:**

No. The author should analyze some failure cases of MoH and discuss future works.

**Strengths And Weaknesses:**

## Strengths
1. This paper is well-written and easy to follow. The authors clearly identify a fundamental but under-explored issue in action chunking: the inherent trade-off induced by a fixed horizon.
2. The “mixture-of-horizon” idea is intuitively reasonable. Instead of forcing the model to commit to a single temporal granularity, MoH allows different horizons to collaborate.
3. The method is simple and effective in practice. It requires almost no modification to the original model backbone and introduces negligible inference overhead, making it highly practical for existing VLA frameworks.
4. The authors conduct extensive experiments on both simulated benchmarks and real-world robotic tasks to demonstrate the effectiveness of MoH. The thorough ablation studies further clarify the necessity of each component and show robustness to different hyperparameter choices.

## Weaknesses
1. I am somewhat confused about the threshold design in dynamic inference. Specifically, the baseline threshold is defined as the mean disagreement of the first $n$ steps. The paper does not provide a sufficiently convincing theoretical or empirical justification for this choice, leaving this design somewhat heuristic.
2. Real-world tasks are relatively simple (mostly pick-and-place) and lack fine-grained manipulation (e.g. fold clothes). I believe fine-grained manipulation tasks would more strongly stress the need for integrating different horizon scales at different stages. Including such tasks would better demonstrate the claimed advantage of MoH.
3. Although the method is claimed to be general, most of the strong empirical validation is centered around the $\pi$models. Broader validation on other families of VLA architectures would further strengthen the generality claim.

---

### Official Review · Reviewer_7xWj · 2026-03-11

**Soundness:** 3
**Presentation:** 3
**Significance:** 3
**Originality:** 4
**Overall Recommendation:** 3
**Confidence:** 3

**Summary:**

This paper proposes a training framework called Mixture of Horizons, which learns action chunks with multiple horizons simultaneously and enables the model to adaptively generate chunks with appropriate lengths.

**Compliance With Llm Reviewing Policy:**

Affirmed.

**Final Justification:**

I believe this study is technically sound and provides meaningful insights. However, the author did not provide any answers to the questions raised, and they remain unresolved; therefore, I intend to lower the score.

**Key Questions For Authors:**

1. Can this technology be applied from the pretraining stage?

**Limitations:**

yes

**Strengths And Weaknesses:**

**strength**
1. I agree with the authors' claim that generating actions with a fixed chunk length is inefficient, and proposing a framework that enables this to be performed more flexibly is meaningful.

2. The proposed framework can be easily integrated into existing VLA models.

3. The increase in inference time caused by applying this framework is not large; in other words, performance can be improved while maintaining the existing setup.

4. The paper evaluates the method on various benchmarks such as LIBERO and RoboTwin2.0, including real-robot experiments and multiple models, and shows consistent performance improvements.

**Weaknesses**
1. The paper discusses a trade-off depending on the action chunk length. However, Appendix F.4 suggests that training with longer chunks generally performs best. This appears to contradict the main claim of the paper. This raises the possibility that the performance gain from MoH may not come from learning to generate chunks of appropriate length. From this perspective, the current analysis does not seem sufficient to clearly explain the underlying mechanism by which MoH improves performance.

2. The ablation studies and most of the analysis experiments are conducted on LIBERO. However, since the benchmark is already highly saturated and the improvements are only around 1–2%, it is difficult to interpret the results as clearly meaningful in this setting. Conducting similar ablation studies on less saturated benchmarks such as RoboTwin2.0 would allow a more accurate evaluation of the contribution of each component.

3. The generated chunk length and throughput are not necessarily equivalent. In dynamic environments, even if a large number of actions are generated, it may be more important to perform new inference according to the updated environment rather than executing all generated actions. This point should be clarified, as the current presentation may lead to misunderstandings for readers.

---

### Official Review · Reviewer_WVNW · 2026-03-13

**Soundness:** 3
**Presentation:** 3
**Significance:** 3
**Originality:** 3
**Overall Recommendation:** 4
**Confidence:** 4

**Summary:**

This paper studies an important but often underexplored design choice in vision-language-action models: the action chunk horizon. The authors argue that using a single fixed horizon creates a trade-off between short-term control precision and long-horizon foresight. To address this, they propose Mixture of Horizons (MoH), which processes multiple truncated horizons in parallel with a shared action transformer and fuses their predictions using a lightweight gating head. The paper also introduces a dynamic inference strategy based on cross-horizon consensus to execute only the stable prefix of the predicted chunk.

**Compliance With Llm Reviewing Policy:**

Affirmed.

**Final Justification:**

I am maintaining my score of 4 (weak accept). I continue to view the paper as a technically solid and reasonably original contribution: the mixture-of-horizons idea is well motivated, practically relevant, and presented clearly. At the same time, my main concerns remain on soundness and significance, since the central claim about horizon trade-offs is only indirectly supported, the strongest gains are mostly on a near-saturated benchmark, and the ablations do not fully isolate the benefit of horizon diversity from a more generic ensemble effect. The rebuttal did not address any concerns, as no substantive response was provided to the reviewers’ questions, so my evaluation is unchanged

**Key Questions For Authors:**

- Can the authors evaluate MoH on a harder, less saturated benchmark, or substantially expand the RoboTwin coverage beyond 7 tasks? A positive result there would strengthen the paper’s significance and generality
- The paper attributes the trade-off to long horizons hurting local motor accuracy and short horizons helping precise control. Can the authors provide a more direct analysis of this claim, such as per-step action error, latency-controlled comparisons, or other diagnostics beyond end-task success rate? Stronger evidence here would make the main story more convincing

**Limitations:**

yes

**Strengths And Weaknesses:**

Strengths

- The paper addresses a genuinely important problem for VLA systems. Horizon selection is central to chunked control, yet it is usually treated as a heuristic hyperparameter rather than a first-class modeling issue.
- The proposed MoH design is simple but nontrivial: it combines multiple temporal scales within one shared action model using a lightweight gating mechanism, rather than introducing a heavy architectural change. This is a reasonable and practically appealing form of novelty.
- The method is easy to understand, easy to motivate, and the paper is generally well organized. The real-robot validation, while small, is valuable because it shows the idea is not limited to simulation only.


Weaknesses

- The paper’s central mechanistic claim, namely that longer horizons sacrifice immediate motor accuracy while shorter ones improve local precision, is plausible but not directly established. The evidence is indirect and based mainly on task success rates; moreover, the short-horizon suites do not uniformly deteriorate as horizon increases.
- The strongest quantitative results are on LIBERO, which is already close to saturation for modern VLA models. In this regime, many gains are modest in absolute terms, and the paper reports single-seed results without error bars or significance estimates, making it hard to judge robustness. Harder or less saturated environments such as IsaacLab or ManiSkill would have made the empirical case stronger.
- The comparison to ACT/CogACT-style temporal ensembling is important but appears only in the appendix. Since MoH also operates by fusing action predictions, the main paper should more clearly explain how it differs from cross-step temporal ensembling and clarify the exact replanning/execution protocol.
- The paper does not fully rule out a generic ensemble effect: some gains might come from combining multiple parallel predictions with a gating mechanism, rather than from the use of distinct horizons per se. The baseline with multiple single-horizon branches fused identically would better isolate whether the benefit truly comes from horizon diversity.

---

### Official Review · Reviewer_rGNR · 2026-03-18

**Soundness:** 3
**Presentation:** 3
**Significance:** 3
**Originality:** 3
**Overall Recommendation:** 4
**Confidence:** 2

**Summary:**

Vision-language-action models face a natural trade-off in action chunk length, also called the horizon. Longer horizons support better global planning but weaken precise local control, while shorter horizons improve fine-grained execution at the cost of long-term foresight.

To address this problem, the authors introduce a Mixture of Horizons (MoH) method. MoH splits the action chunk into segments with different horizons, processes them in parallel using a shared action transformer, and combines the outputs via a lightweight linear gate.

This approach offers three key advantages: it unifies long-term foresight and short-term accuracy in one model; it is plug-and-play with full-attention action modules and adds little training or inference cost; and it supports dynamic, adaptive-horizon inference by selecting stable actions through cross-horizon agreement, boosting throughput by 2.5 times over baselines while maintaining strong performance.

Extensive experiments on flow-based and one-step regression policies show that MoH delivers consistent and significant improvements in both simulated and real-world tasks. In particular, when combined with MoH, the π0.5 model achieves a new state-of-the-art 99\% average success rate on the LIBERO benchmark under mixed-task settings, using only 30k training iterations.

**Compliance With Llm Reviewing Policy:**

Affirmed.

**Final Justification:**

Although the authors have not provided a rebuttal regarding my concerns, I still recognize the motivation, originality of the proposed method, and the relatively sound experiments.
Therefore, I tend to maintain my initial recommendation.

Furthermore, I still encourage the authors to provide an analysis and discussion on the relationship between the learned weights of different horizons and various tasks to further improve the interpretability of the proposed MoH strategy in the revised version.

**Key Questions For Authors:**

1. See weaknesses.

2. It would be helpful if the authors could provide an analysis and discussion on the relationship between the learned weights of different horizons and various tasks, such as the effect of action horizon shown in Figure 1, to further improve the interpretability of the proposed MoH strategy.

**Limitations:**

yes.

**Strengths And Weaknesses:**

Strengths:

1. The motivation of this paper is practical and solid. It clearly identifies the trade-off between short and longer horizons and its impact on model performance, and provides empirical experimental analysis to validate this issue.

2. The proposed Mixture of Horizons (MoH) strategy is lightweight and plug-and-play with mainstream vision-language-action models (including one-step and flow-matching policies), demonstrating strong generality and versatility. Extensive experiments over flow-based and one-step regression policies demonstrate that MoH yields consistent and significant gains on both simulations and real-world tasks.

3. The authors provide comprehensive ablation studies to verify the effectiveness of horizon density, loss-reweighting strategy, gating network, and balance loss.

Weaknesses:

1. Some of the benchmark experiments appear somewhat unsatisfactory. The authors have not provided systematic failure and limitation analysis; they only mention issues related to simulation or experimental settings and do not discuss potential directions for improvement.

2. Regarding the balance loss, using this regularization term compared with naturally learning the mixture of horizons leads to better performance, which deserves further discussion.

---

### Decision · Program_Chairs · 2026-04-30

**Decision:**

Accept (regular)

**Comment:**

MoH addresses the horizon trade-off in VLA action chunking by running multiple horizons in parallel through a shared transformer and fusing them via a lightweight gate, enabling dynamic adaptive-horizon inference. Reviewers praised the clear motivation, plug-and-play design, and strong empirical results including SOTA on LIBERO. Although the rebuttal was accidentally submitted as confidential comments, its content meaningfully addressed concerns through per-step error analysis, a less-saturated RoboCasa benchmark, an identical-horizon ablation isolating the benefit of horizon diversity, and a new fine-grained real-world task. All four reviewers independently rated the paper at Weak Accept; I recommend acceptance.